# A Quadruple Gene-Deleted Live BoHV-1 Subunit RVFV Vaccine Vector Reactivates from Latency and Replicates in the TG Neurons of Calves but Is Not Transported to and Shed from Nasal Mucosa

**DOI:** 10.3390/v16091497

**Published:** 2024-09-21

**Authors:** Selvaraj Pavulraj, Rhett W. Stout, Daniel B. Paulsen, Shafiqul I. Chowdhury

**Affiliations:** Louisiana Animal Disease Diagnostic Laboratory, Department of Pathobiological Sciences, School of Veterinary Medicine, Louisiana State University, Baton Rouge, LA 70803, USA; pselvaraj1@lsu.edu (S.P.); rstout1@lsu.edu (R.W.S.); dpauls1@lsu.edu (D.B.P.)

**Keywords:** bovine herpesvirus, BoHV-1, vectored vaccine, latency reactivation, glycoprotein E, quadruple mutant virus, BoHV-1qmv vector, trigeminal ganglion, cattle, DIVA

## Abstract

Bovine herpesvirus type 1 (BoHV-1) establishes lifelong latency in trigeminal ganglionic (TG) neurons following intranasal and ocular infection in cattle. Periodically, the latent virus reactivates in the TG due to stress and is transported anterogradely to nerve endings in the nasal epithelium, where the virus replicates and sheds. Consequently, BoHV-1 is transmitted to susceptible animals and maintained in the cattle population. Modified live BoHV-1 vaccine strains (BoHV-1 MLV) also have a similar latency reactivation. Therefore, they circulate and are maintained in cattle herds. Additionally, they can regain virulence and cause vaccine outbreaks because they mutate and recombine with other circulating field wild-type (wt) strains. Recently, we constructed a BoHV-1 quadruple mutant virus (BoHV-1qmv) that lacks immune evasive properties due to UL49.5 and glycoprotein G (gG) deletions. In addition, it also lacks the gE cytoplasmic tail (gE CT) and Us9 gene sequences designed to make it safe, increase its vaccine efficacy against BoHV-1, and restrict its anterograde neuronal transport noted above. Further, we engineered the BoHV-1qmv-vector to serve as a subunit vaccine against the Rift Valley fever virus (BoHV-1qmv Sub-RVFV) (doi: 10.3390/v15112183). In this study, we determined the latency reactivation and nasal virus shedding properties of BoHV-1qmv (vector) and BoHV-1qmv-vectored subunit RVFV (BoHV-1qmv sub-RVFV) vaccine virus in calves in comparison to the BoHV-1 wild-type (wt) following intranasal inoculation. The real-time PCR results showed that BoHV-1 wt- but not the BoHV-1qmv vector- and BoHV-1qmv Sub-RVFV-inoculated calves shed virus in the nose following dexamethasone-induced latency reactivation; however, like the BoHV-1 wt, both the BoHV-1qmv vector and BoHV-1qmv Sub-RVFV viruses established latency, were reactivated, and replicated in the TG neurons. These results are consistent with the anterograde neurotransport function of the gE CT and Us9 sequences, which are deleted in the BoHV-1qmv and BoHV-1qmv Sub-RVFV.

## 1. Introduction

Bovine herpesvirus type 1 (BoHV-1) causes significant economic losses to the cattle industry worldwide. BoHV-1 infection is also implicated as an initiator or an essential member of the fatal bovine respiratory disease complex (BRDC) caused by several respiratory viruses, i.e., bovine viral diarrhea virus (BVDV), bovine respiratory syncytial virus (BRSV) and bacteria, i.e., *Mannheimia hemolytica* [1,2,3]. Initially, BoHV-1 replicates in the nasal epithelium following intranasal infection and causes rhinotracheitis, conjunctivitis, abortion, and fertility problems in cattle [4]. Subsequently, the virus particles enter the sensory nerve endings of the trigeminal nerve in the nasal mucosa, and the viral nucleocapsid with a few tegument proteins move along the axon retrogradely to the neuron cell bodies in the trigeminal ganglia (TG). Once in the TG, the virus has limited replication and eventually establishes life-long latency by maintaining the viral genome as an episome in the nucleus of TG neurons [5]. The viral genome does not replicate during latency, but a latency-related gene (LRG) is transcribed [6]. Upon stress or the systemic administration of synthetic corticosteroid dexamethasone (Dex), immediate early and early gene transcriptions/expressions ensue, followed by viral genome replication, late viral protein synthesis, and assembly in the TG [7]. Subsequently, mature virus particles are transported anterogradely from the neuron cell bodies in the TG to the nerve endings in the nasal mucosa—where the virus replicates again, resulting in nasal virus shedding [8]. Notably, animals shed the virus in the nose during latency reactivation with or without clinical signs, and that is how the latent BoHV-1 wild-type carrier becomes a potential source for virus transmission to naïve animals in cattle herds [9]. Importantly, subclinical secondary intranasal, endogenous BoHV-1 infection also causes immunosuppression, which facilitates *M. hemolytica* colonization, resulting in secondary bacterial fatal pleuropneumonia or other respiratory viral infections (BVDV and BRSV). Collectively, these infections are defined as bovine respiratory disease complex (BRD) [10].

Like the wild-type (wt) BoHV-1, traditional BoHV-1 modified-live vaccine viruses (MLVs) also have similar latency reactivation, immune suppression, and nasal virus-shedding properties following intranasal vaccination [11]. Therefore, they also circulate and are maintained in the cattle population. Consequently, they have the potential to mutate and recombine with circulating field viruses noted above, resulting in the emergence of revertant or variants that cannot be definitively linked to the vaccine strain because MLVs lack serologically distinguishable marker or DIVA properties (differentiating infected animals from the vaccinated animals serologically) [12]. Further, reactivated MLVs have been suspected to be the cause of the outbreaks of infectious bovine rhinotracheitis (IBR) and abortion and fertility issues in pregnant cows and heifers in the field [13]. However, the source of these viruses is not clear because MLVs lack the serological marker and are indistinguishable from the wt field strains. To address these problems associated with BoHV-1 MLVs, they are banned in EU countries. Instead, the BoHV-1 glycoprotein E (gE)-deleted DIVA or marker vaccine is the only mandated vaccine that is allowed. Using this strategy, BoHV-1 was eradicated in most EU and a few non-EU countries [14]. Based on our previous results, the gE-deleted marker vaccine was highly attenuated, and under experimental conditions, the gE-deleted virus established latency in the TG neurons following intranasal conditions and did not shed the virus in the nasal secretions upon dexamethasone-induced latency reactivation. However, in 2015, the gE-deleted vaccine virus was isolated in the nasal swabs of cows eight months after vaccination with the gE-deleted marker vaccine [15]. As noted above, the gE-deleted vaccine can also establish latency in the TG. While it is unclear whether these vaccine isolates were the latent vaccine virus that was reactivated, the circumstances surrounding this vaccine virus isolation indicated that was the case. Apparently, there were no other sources of BoHV-1 infection, and no vaccination had been performed on the farm during the previous four months. Therefore, only virus reactivation and excretion by cattle on the farm could explain the source of gE-negative BoHV-1 in the cows’ nasal secretions [15]. Two BoHV-1 envelope protein gEs, specifically gE cytoplasmic tail (CT) and Us9, are crucial for the anterograde axonal transport of the virus from the cell bodies in the TG neurons to their processes in the nose. As a result, the individual gE, gE CT- or Us9 deletion mutants are not shed from the nose following latency reactivation [16,17,18,19,20].

To improve the safety and efficacy of the BoHV-1 gE-deleted marker vaccine, we combined gE CT and entire Us9 ORF deletions in the backbone of the BoHV- 1 U_L_49.5 mutant, in which major histocompatibility complex (MHC) class I down-regulation domains are deleted [21]. The resulting BoHV-1 triple mutant virus (BoHV-1tmv) was not characterized concerning its latency reactivation property. Since calves latently infected either with gE CT or the Us9 deletion mutant did not shed the virus in the nose following dexamethasone (Dex)-induced latency reactivation [20,22], we assumed that the triple gene mutant BoHV-1 (BoHV-1tmv) containing simultaneous gE CT and Us9 deletions would not shed in the nose following Dex-induced reactivation. To further improve the immunogenicity of the BoHV-1tmv, we deleted the BoHV-1 chemokine-binding envelope glycoprotein gG of BoHV-1tmv, resulting in BoHV-1qmv [23]. Subsequently, we engineered the BoHV-1qmv to serve as a subunit vaccine vector for bovine viral diarrhea virus type 2 (BoHV-1qmv Sub-BVDV2) [24] and Rift Valley fever virus (BoHV-1qmv Sub-RVFV) [25] and determined that the BVDV subunit vaccines protected the vaccinated calves against the BVDV challenge. Additionally, we determined that the BoHV-1qmv Sub-RVFV induced RVFV (MP12)-specific neutralizing antibody and cellular immune responses [25].

In the present study, our goal was to validate that both BoHV-1qmv and BoHV-1qmv Sub-RVFV viruses establish latency in the TG neurons, but upon Dex-induced latency reactivation, they do not shed in the nose. The results presented demonstrate that, like BoHV-1 wt, both BoHV-1qmv vector and BoHV-1qmv Sub-RVFV vaccine viruses established latency in the TG neurons, were reactivated and replicated in the TG neurons upon Dex-induced reactivation. However, only the BoHV-1 wt virus is transported to the nerve endings in the nasal mucosa and shed in the nasal secretions of the infected calves.

## 2. Materials and Methods

### 2.1. Ethical Statement

The study was performed following protocols approved by the LSU Institutional Animal Care and Use Committee, Louisiana State University (IACUC Protocol #20-028).

### 2.2. Cells and Medium

The bovine esophagus cells (KOP-R cells), 293T and the Madin Darby bovine kidney (MDBK) cell lines were maintained in Dulbecco’s modified Eagle medium (DMEM #10-017-CV, Corning^®^, Corning, NY, USA) supplemented with 10% heat-inactivated EquaFETAL serum (Atlas Biologicals, Fort Collins, CO, USA) and 1× antibiotic/antimycotic solution (#30-004-CI; Corning^®^).

### 2.3. Viruses

BoHV-1 wt Cooper (Colorado-1) strain was obtained from the American Type Culture Collection (#VR-864, ATCC^®^, Manassas, VA, USA), and low-passage viral stocks were maintained at −80 °C. The recombinant BoHV-1qmv vector virus and BoHV-1qmv Sub-RVFV vaccine virus were constructed and characterized previously [24]. The BoHV-1 wt and the BoHV-1 recombinant viruses were titrated by plaque assay in MDBK cells, as described previously [24].

### 2.4. Antibodies

BoHV-1 glycoprotein C (gC)-specific monoclonal antibody (F2) was purchased from the Monoclonal Antibody Center, the Department of Veterinary Microbiology and Pathology, Washington State University, Pullman, WA. Donkey anti-mouse IgG Alexa Fluor 488 (#A-21202) was purchased from Thermo Fisher scientific^®^ (Waltham, MA, USA).

### 2.5. Animal Experiment

Seventeen six-month-old crossbred steers, bulls, or heifer non-vaccinated calves were obtained from a BVDV-free supplier. The calves were pre-tested for BoHV–1 serum neutralizing (SN) antibody titers and BVDV-viremia (VetMAX™-Gold BVDV PI Detection Kit, #4413938, Thermo Fisher Scientific, Plaquemine, LA, USA) to ensure BoHV–1/BVDV-free status. BVDV-negative calves with <4 BoHV-1-specific SN antibody titers were selected for the study. Calves were randomly divided into four groups. Group 1 (negative control; mock-infected) consisted of one calf. Group 2 (BoHV-1 wt group) consisted of five calves, group 3 (BoHV-1qmv vector control) consisted of three calves, and group 4 (BoHV-1qmv Sub-RVFV) consisted of eight calves (Figure 1). The calves in BoHV-1 wt were housed in a different barn (a swine barn). BoHV-1qmv vector group and BoHV-1qmv Sub-RVFV group calves were housed in the LSU-pole barn but separated by 200 feet. One negative control calf in the pole barn was kept separately (100 feet apart) from the two vaccine group calves.

Virus inoculation, Dex-induced latency reactivation, and sample collection schemes are shown in Figure 1. After a week of acclimatization, the negative control group (group 1) was mock-infected intranasally (IN) with a cell culture medium. Five calves in the BoHV-1 wt group (group 2) were inoculated IN with 1 × 10^7^ PFU/nostril of BoHV-1wt (a total of 2 × 10^7^ PFUs per animal). Three calves in the BoHV-1qmv vector group (group 3) were similarly inoculated with BoHV-1qmv 1 × 10^8^ PFU/nostril (a total of 2 × 10^8^ PFUs per animal) and subcutaneously (SQ) with 1 × 10^8^ PFU. Eight calves in the BoHV-1qmv Sub-RVFV vaccine group (group 4) were given IN with BoHV-1qmv Sub-RVFV 4 × 10^7^ PFU/nostril (a total of 8 × 10^7^ PFUs per animal) and SQ with 4 × 10^7^ PFU. Blood for serum and nasal swabs were collected on the indicated days (Figure 1). At 28 days post infection (dpi), the negative control calf, two calves in group 2 (wt), one calf in group 3 (vector), and four calves in group 4 (subunit vaccine) were euthanized (with Euthasol^®^-Euthanasia Solution; pentobarbital sodium and phenytoin sodium). The remaining calves in each group were inoculated intravenously (I/V) with dexamethasone (5 mg/kg), followed by two more S/C Dex injections (0.25 mg/kg) at 29 and 30 dpi as described previously [22] to reactivate the corresponding latent viruses. The experiment was terminated at 33 dpi, calves were euthanized as above, and their TGs were collected.

### 2.6. Clinical Parameter and Sample Collection

For clinical signs, rectal temperatures, nasal discharge, and nasal lesions were recorded during each visit, and clinical scores were assigned for each parameter. Rectal temperatures were scored 0–4 (<39.0 °C, 39.5 °C, 40.0 °C, 40.5◦C and >40.9 °C), nasal discharges were scored 0–4 (normal, serous, mild and severe mucopurulent), and nasal lesions were scored 0–3 (normal, hyperemia, pustules and ulcers).

Nasal swabs were collected in 2 mL of DMEM, supplemented with 3× antibiotic–antimycotic solution and 2% FBS. Nasal swab samples were aliquoted and stored at −80 °C until use. Blood samples collected for sera were processed, aliquoted, and stored at −80 °C.

### 2.7. Necropsy and Sample Collection

At day five post-dexamethasone treatment (5 dp-Dex), calves were sedated by xylazine and euthanized with Euthasol^®^ (euthanasia solution; pentobarbital sodium and phenytoin sodium). TGs were collected at necropsy for histopathology (10% formalin), virus isolation, and qPCR assays (dry ice). Formalin-fixed tissues were paraffinized, sectioned, and processed for histopathology (H&E staining).

### 2.8. Serum Virus Neutralization Assay

BoHV-1-specific neutralizing antibody titers in sera were determined by a standard plaque reduction assay (50% reduction/neutralization of the approx. 100 PFUs) as described previously [24]. Viral plaques were counted 72 h post infection (hpi) after the cells were fixed (4% paraformaldehyde) and stained (0.1% crystal violet).

### 2.9. DNA/RNA Isolation, cDNA Synthesis, and BoHV-1-Specific Quantitative PCR (q-PCR)

To determine the BoHV-1 genome copies in the nasal swabs and the TG, total DNA was isolated using the QIAamp^®^ DNA mini kit (#51306, Qiagen, Hilden, North Rhine-Westphalia, Germany). Before DNA isolation from TG, tissues were homogenized using 2.8 mm ceramic beads (#15-340-154, Thermo Fisher Scientific^®^, Waltham, MA, USA) in Pre-cellys 24 homogenizers (#13112, Bertin Instruments, Rockville, MD, USA). RNA was isolated from the TG samples to detect and quantify the BoHV-1- latency-related, immediate early protein (infected cell protein 0; ICP0), and glycoprotein C (true late protein). In brief, after the homogenization, RNA was isolated using the RNeasy mini kit (#74104, Qiagen). DNA contamination in RNA samples was removed by the RNase-Free DNase set (#79254, Qiagen). The cDNA was synthesized using the Verso cDNA synthesis kit (AB-1453/A, Thermo Fisher Scientific^®^). Targeted gene/transcript copies were determined by TaqMan probe-based real-time qPCR in the ABI PRISM™ 7900HT Sequence Detection System (Applied Biosystems, Waltham, MA, USA), using the specific primers and probes given in Table 1. Each time, the PCR reaction setup was run with six standards of known quantity (10^1^ to 10^6^ copies per reaction). BoHV-1 genome copies in the nasal samples (normalized to 100 ng of total DNA) were compared with the generated standard curves. The assay was duplicated, and the results were expressed as BoHV-1 genome copies per 100 ng of DNA. The BoHV-1 genome copies in the TGs were calculated by normalizing the MCP-specific CT values against the standard curve generated based on the CT values obtained for the known housekeeping gene, glyceraldehyde 3-phosphate dehydrogenase (GAPDH) copies (two copies/cell), in the same cells. The mean copies of the BoHV-1-MCP gene per one million cells were then plotted. The qPCR gene transcript copy results were expressed as transcript copies per ng of RNA.

### 2.10. Indirect Immunofluorescence Assay

To confirm the active virus replication in the TG neurons, DNA replication-dependent, true late gC gene [26,27] expression was determined by an indirect immunofluorescence assay (IFA) using a gC-specific mAb, F2 [28]. Briefly, deparaffinized and rehydrated TG tissue sections containing slides were soaked in a working citrate buffer (9 mL of 0.1 M citric acid + 41 mL of 0.1 M sodium citrate in 500 mL with water) and microwaved for 5 min for the antigen retravel. After cooling at room temperature for 20 min, they were washed three times in water, followed by permeabilization with 0.2% *v*/*v* triton-100 in tris-base saline, with 10 min of incubation twice at room temperature. Subsequently, tissue sections were blocked with 3% bovine serum albumin (BSA) at 37 °C for one hour. Primary and secondary antibodies were diluted in 0.3% BSA and incubated successively at 37 °C for one hour each. Finally, slides were washed, counterstained with DAPI, mounted with 50% *v*/*v* glycerol in PBS, and examined under a fluorescent microscope.

### 2.11. Statistical Analysis

Statistical analyses were performed using GraphPad PRISM^®^ 5.01 software (San Diego, CA, USA). The Shapiro–Wilk test was performed to determine the normal distribution of the data. Normally distributed data were analyzed by two-way ANOVA followed by Bonferroni post tests to compare replicate means by row. A ‘*p*’ value of less than 0.05 was considered significant for all analyses.

## 3. Results

### 3.1. Clinical Signs and Nasal Virus Shedding of BoHV-1qmv Sub-RVFV- and BoHV-1qmv Vector-Infected/Vaccinated Calves Compared with the BoHV-1 wild-type (wt)-Infected Calves

Following intranasal infection/vaccination, BoHV-1qmv- and BoHV-1qmv Sub-RVFV-inoculated calves remained clinically normal, with low clinical scores (Figure 2; Appendix A). On the contrary, from 5 to 7 dpi, BoHV-1 wt-infected calves developed a high fever, nasal discharge, and lesions in the nasal mucosa typical for BoHV-1 wt infection, resulting in clinical scores ranging between 6 and 6.6 (Figure 2).

### 3.2. Nasal Virus Shedding following Intranasal (IN) Infection of BoHV-1 wt, BoHV-1qmv Vector, and BoHV-1qmv Sub-RVFV

BoHV-1qmv and BoHV-1qmv Sub-RVFV replicated moderately in the nasal epithelium until three dpi and shed viruses in the nose ranging between 1 and 2 × 10^3^ PFUs, respectively. By 5 dpi, the quantities of viruses shed were reduced to 0.8 × 10^2^ PFUs (BoHV-1qmv) and 0.6 × 10^1^ PFUs (BoHV-1qmv Sub-RVFV), respectively (Figure 3; Appendix A). In comparison, the BoHV-1 wild-type replicated with a significantly higher yield on both days, 6.6 × 10^4^ (3 dpi) PFUs and 8.1 × 10^5^ PFUs (5 dpi), respectively, which declined to 3 × 10^1^ PFUs by 14 dpi. Therefore, even when compared to the amounts of BoHV-1 wt virus, 10-fold greater BoHV-1qmv and 4-fold greater BoHV-1qmv Sub-RVFV were inoculated intranasally (Figure 1), and the amounts of virus shed in the nose of BoHV-1qmv- and BoHV-1qmv Sub-RVFV-infected calves were reduced 30-fold (BoHV-1qmv) and 60-fold (BoHV-1qmv Sub-RVFV) at 3 dpi. At 5 dpi, BoHV-1qmv and BoHV-1qmv Sub-RVFV viruses shed were reduced yet 1000- and 1300-fold relative to that of BoHV-1 wt. These results indicated that BoHV-1qmv and BoHV-1qmv Sub-RVFV were highly attenuated. From 21 dpi and through 28 dpi, no virus was detectable in the nasal swabs of all three viruses, even with a sensitive qPCR test, which confirmed their latency (Figure 4; Appendix A).

### 3.3. Following Dex-Induced Reactivation, Only the BoHV-1 wt-Infected Calves Shed the Virus in the Nose

At 28 dpi, one sentinel/negative control calf, two BoHV-1 wt-, one BoHV-1qmv-, and four BoHV-1qmv Sub-RVFV- infected calves were euthanized to determine the latency in the TG neurons. At the same time, all the remaining calves were injected intravenously and subcutaneously for two consecutive days with dexamethasone to reactivate the latent virus. As shown in Figure 3 and Appendix A, all three calves in the wt group shed the virus in the nasal swab at five dp-Dex. In contrast, none of the calves in the BoHV-1qmv vector or BoHV-1qmv Sub-RVFV vaccine group shed virus in the nasal secretions upon Dex treatments. These results were also validated by highly sensitive qPCR analysis, which detected a mean of 4.8 × 10^3^ genome copies in the nasal swabs collected at five dp-Dex reactivation from calves infected with BoHV-1 wt. Still, none could be detected for the BoHV-1qmv- and BoHV-1qmv Sub-RVFV-infected calves (Figure 4 and Appendix A). These results imply that the BoHV-1 wt virus was reactivated and transported anterogradely from the TG to the nasal epithelium but not BoHV-1qmv and BoHV-1qmv Sub-RVFV.

### 3.4. Like BoHV-1 wt, BoHV-1qmv, and BoHV-1qmv Sub-RVFV Established Latency, Was Reactivated from Latency, Expressed Immediate Early Protein 0 (ICP0), and DNA Replication-Dependent True Late Envelope Protein gC in the TG Neurons but Unlike the BoHV-1 wt Was Not Transported from the TG to and Shed in the Nose

During alpha herpesvirus latency in the peripheral sensory neurons, the linear viral genome circularizes upon entry into the nucleus of the TG neuron, and the episomal, circular and latent viral genome is not replicated. Importantly, only the latency-related gene (LRG) is transcribed [6]. As shown in Figure 5 and Appendix A, qPCR analysis detected a mean of 9744 genome copies per million cells in the BoHV-1 wt latently infected TGs. In comparison, 756 and 925 BoHV-1 genome copies were detected in the TGs of calves infected with BoHV-1qmv and BoHV-1qmv Sub-RVFV, approximately 13-fold and 10-fold less compared with BoHV-1 wt, respectively. However, at five dp-Dex, the mean genome copies for BoHV-1 wt, BoHV-1qmv, and BoHV-1qmv sub-RVFV were 145,915, 7288, and 8030, respectively, which were 15-, 9-, and 8-fold higher compared to their genome copy numbers in the latent TG samples. Therefore, all three viruses were replicated in the TG neurons upon Dex-induced reactivation.

As shown in Figure 6A and Appendix A, LRG transcripts were detected in the TGs of all three groups of calves during latency regardless of the virus groups. Specifically, in calves infected with BoHV-1 wt, BoHV-1qmv vector, and BoHV-1qmv Sub-RVFV, the mean copy numbers of LRG transcripts detected per ng of TG RNA were 1944, 102, and 100, respectively. Therefore, relative to the BoHV-1-wt LRG transcript copy numbers, the corresponding values for BoHV-1qmv and BoHV-1qmv-RVFV were reduced by approximately 19-fold, which is discussed later.

During alpha herpesvirus productive replication, immediate early genes are transcribed first, followed sequentially by immediate early protein translation. Subsequently, early gene transcription and translation occur, followed by DNA replication and late gene transcription and translation [7]. Among the late genes, certain gene transcriptions, such as gC, occur only from the replicated DNA and therefore are DNA replication-dependent true late genes, and their transcription and translation are increased upon DNA replication [27]. Therefore, detecting immediate early protein ICP0 upon latency reactivation indicates that the latent virus has reactivated. Similarly, detecting the true late gene, envelope glycoprotein gC transcripts, or ion or protein indicates that the viral genome has replicated upon reactivation.

As shown in Figure 6B and Appendix A, at five dp-Dex, a mean of 2700 copies, 77 copies, and 110 copies of ICP0 transcripts/ng of TG RNA were detected from the BoHV-1 wt-, BoHV-1qmv-, and BoHV-1qmv Sub-RVFV-infected calves, respectively, upon latency reactivation. On the contrary, ICP0 transcripts were not detectable in the similarly processed TG samples of the corresponding TG samples collected during the latency. Therefore, all three viruses, BoHV-1wt-, BoHV-1qmv-, and BoHV-1qmv Sub-RVFV-infected calves, were reactivated from latency, although the ICP0 copy numbers for the two recombinant viruses were 35-fold (BoHV-1qmv) and 25-fold (BoHV-1qmv Sub-RVFV) reduced compared to that of BoHV-1 wt).

To confirm that following the latency reactivation, the BoHV-1 wt, BoHV-1qmv vector, and BoHV-1qmv Sub-RVFV viruses replicated in the TG neurons, we investigated the expression of late BoHV-1 protein gC in the TG during latency and at five dp-Dex by qPCR targeting the gC transcript and IFA targeting gC using anti-gC Mabs (F2). The qPCR (Figure 6C and Appendix A) revealed that at five dp-Dex, a mean of 47,051 copies, 661 copies, and 720 copies of gC transcripts/ng of TG RNA were detected from the BoHV-1 wt-, BoHV-1qmv-, and BoHV-1qmv Sub-RVFV-infected calves, respectively, upon latency reactivation. On the contrary, gC transcripts were not detectable in the similarly processed corresponding TG samples collected during the latency. Therefore, it validates our ICP0 transcript expression profiles observed in the latency-reactivated calves. Further, the IFA results shown in Figure 7A,B confirm that regardless of BoHV-1 wt-, BoHV-1qmv vector- or BoHV-1qmv Sub-RVFV-infected calves, gC is expressed in a few TG neurons at 5 days post reactivation but not in their corresponding latent TG neurons. However, the number of TG neurons expressing gC was higher in the case of wt-infected calves compared with that of the BoHV-1qmv- and BoHV-1qmv Sub-RVFV-infected calves. Taken together, the results of nasal virus shedding, ICP0 transcription, DNA replication, and gC transcription/expression in the TG neurons of calves following Dex-induced reactivation suggest that, upon latency reactivation, both BoHV-1qmv and BoHV-1qmv Sub-RVFV viruses were reactivated and replicated and most likely assembled into mature viral particles, as in the case of the wild-type.

### 3.5. Infiltration of Inflammatory Cells in the TGs of the BoHV-1 wt-Infected Calves during Latency and upon Reactivation but Not in TGs of BoHV-1qmv- and BoHV-1qmv Sub-RVFV-Infected Calves

The TG from the BoHV-1 wt-infected latent calves showed small foci of inflammation in the neuropil in the brain (Figure 8B), along with a few scattered individual inflammatory cells in the remainder of the neuropil. The inflammatory infiltration mostly comprised plasma cells, lymphocytes, and microglial cells. In contrast, none or negligible inflammatory infiltrates were visible in the TGs of the latently infected BoHV-1qmv-(vector)- and BoHV-1qmv Sub-RVFV-vaccinated calves (Figure 8D and F, respectively). Upon Dex-induced reactivation, TGs from the wt-infected calves revealed even further increased infiltrates in the TGs of the BoHV-1 wt group (Figure 8C). However, the histopathology of the Dex-reactivated TG sections of BoHV-1qmv- and BoHV-1qmv Sub-RVFV-vaccinated calves (Figure 8E and G, respectively) were very similar to their corresponding latently infected TGs (Figure 8D and F, respectively) or the TG of a control uninfected calf (Figure 8A).

### 3.6. Dexamethasone-Induced Latency Reactivation Resulted in Memory-Neutralizing Antibody Response Only for BoHV-1 wt-Infected Calves but Not for the BoHV-1qmv- and BoHV-1qmv Sub-RVFV-Infected Calves, Validating Another Way of BoHV-1 wt Reactivation in and Transport from the TG to Nasal Mucosa, Followed by Replication and Shedding in the Nasal Mucosa

As shown in Figure 9 and Appendix A, at latency (28 dpi), the average SN titer in the BoHV-1 wt, BoHV-1qmv vector, and BoHV-1qmv Sub-RVFV group calves were 70.1, 38.1, and 17.4, respectively. At five dp-Dex, the mean SN titer of the BoHV-1 wt-infected calves rose from 70.1 to 596.2 (approx. 19-fold). On the contrary, the corresponding mean SN titers of both BoHV-1qmv vector- and BoHV-1qmv Sub-RVFV-infected calves dropped slightly compared to that at 0 dp-Dex (28 dpi) from 38.1 to 18.7 (BoHV-1qmv) and from 17.4 to 16.92 (BoHV-1 Sub-RVFV). These results reinforce that only the BoHV-1 wt shed virus in the nose at 5 dp-Dex. Therefore, BoHV-1wt is transported from the TG neurons to the nasal mucosa and replicated there, stimulating the memory immune response. However, both BoHV-1qmv and BoHV-1qmv Sub-RVFV were not transported to the nasal mucosa, and their SN titers dropped slightly instead.

## 4. Discussion

We have constructed the BoHV-1qmv vector virus by deleting and mutating four viral gene sequences, gE-CT, US9, UL49.5, and gG. BoHV-1qmv is highly attenuated because it lacks immuno-suppressive and anterograde neuronal transport properties [16,17,21,24]. We have used this BoHV-1qmv vector to generate a subunit vaccine against RVFV [25]. In the current investigation, the results demonstrated that, like the BoHV-1 wt, BoHV-1qmv vector, and BoHV-1qmv Sub-RVFV established latency in the TG, were reactivated from latency and replicated in the TG neurons upon the Dex treatments, although at a significantly reduced level. However, only BoHV-1 wt, not the BoHV-1qmv vector, and BoHV-1qmv Sub-RVFV shed virus in the nasal secretions.

Previously, we reported that BoHV-1 mutants lacking either gE-CT or Us9 were not transported anterogradely from the neuronal cell bodies to the axon termini in rabbit primary neuronal cultures. Therefore, the lack of nasal viral shedding in BoHV-1qmv- and BoHV-1qmv Sub-RVFV-infected calves following latency reactivation could only be due to the gE-CT and Us9 deletions required for the anterograde axonal transport from the neuron cell bodies in the TG to axon termini in the nasal mucosa [20,22]. We validated the results in two ways; first, we could not detect any BoHV-1qmv and BoHV-1qmv Sub-RVFV-specific genomic DNA in the noses of the calves following latency reactivation. Meanwhile, the BoHV-1 wt virus and its genomic DNA were readily detected. Second, consistent with these findings, there was a BoHV-1-specific memory-neutralizing antibody response in the BoHV-1 wt-infected calves following the Dex-induced reactivation. In the BoHV-1 wt-infected calves, the virus was transported to the nasal mucosa and replicated, resulting in a boosting effect. This was not the case for both BoHV-1qmv and BoHV-1qmv Sub-RVFV. The results of nasal virus shedding following wt, BoHV-1qmv, and BoHV-1qmv Sub-RVFV infection in calves show that even though we inoculated 10-fold more viruses in each case compared to that of wt, the amount of mutant virus shed was 30-60-fold. Also, in the TG neurons, the amount of viral genomic DNA copies detected at latency and at five dp-Dex for both BoHV-1qmv and BoHV-1qmv Sub-RVFV were reduced proportionately. For the same reason, the copy numbers of the ICP0, LAT, and gC transcripts are significantly reduced compared to the wt. Therefore, these results are consistent with the significant attenuation of the prototype vaccine virus. Based on the neutralizing titers following reactivation, there was about a 19-fold increase in the BoHV-1 antibody titer in the wt group due to the memory immune response. This is because BoHV-1 wt replicated in the nose, resulting in a boosting effect. BoHV-1qmv and BoHV-1qmv Sub-RVFV were not transported to the nose; hence, no memory immune response was observed. Nevertheless, if the BoHV-1qmv- and BoHV-1qmv Sub-RVFV-vaccinated animals were to obtain a booster vaccination, they would have had a memory immune response against the vector and the subunit RVFV antigen.

Both BoHV-1qmv and BoHV-1qmv Sub-RVFV replicated in the TG neurons following the Dex-induced reactivation, even though they did not shed in the nose. To ensure that the vaccine virus does not even replicate in the TG neurons, deleting the viral thymidine kinase (TK) gene would be necessary because the neurons are post-mitotic and do not express TK. Previously, we have constructed a triple gene mutant pseudorabies (PRVtmv)-vectored subunit vaccine against porcine circovirus type 2b (PCV2b) and classical swine fever virus (CSFV) (PRVtmv+) in which, in addition to the gE and gG deletions, we deleted the thymidine kinase (TK) gene. PRVtmv+ replicated in the epithelial cells in vitro and in the nasal mucosa of pigs in vivo because the cellular TK complemented the viral TK. However, as expected, PRVtmv+ did not replicate in the TG neurons following the Dex-induced reactivation of the latent PRVtmv+ and was not shed in the nose [29]. Therefore, we plan to delete the TK gene of the BoHV-1qmv vector and make it even safer.

## 5. Conclusions

BoHV-1qmv is safe as a subunit live viral vector and does not shed in nasal secretions upon latency reactivation. This property of the BoHV-1qmv vector prevents the likelihood of vaccine virus circulation in cattle and the risk of reversion to virulence. Therefore, the BoHV-1qmv can be used as a safe subunit vaccine vector for various diseases of cattle.

## Figures and Tables

**Figure 1 viruses-16-01497-f001:**
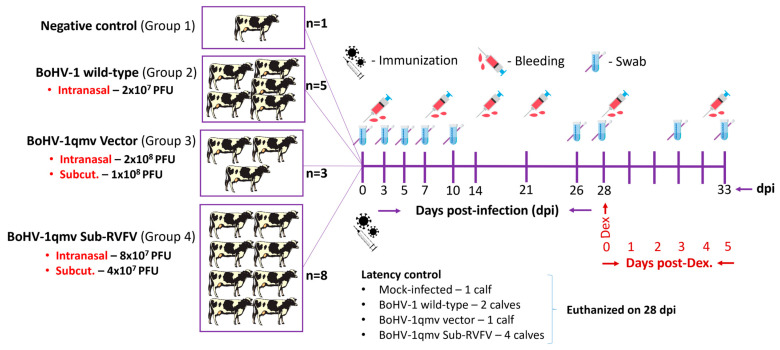
Infection, sample collection, dexamethasone-induced latency reaction, and euthanasia scheme for the animal experiment. PFUs—plaque-forming units; Subcut.—subcutaneous injection; Dex.—dexamethasone; Days post-Dex.—Days post-dexamethasone treatment.

**Figure 2 viruses-16-01497-f002:**
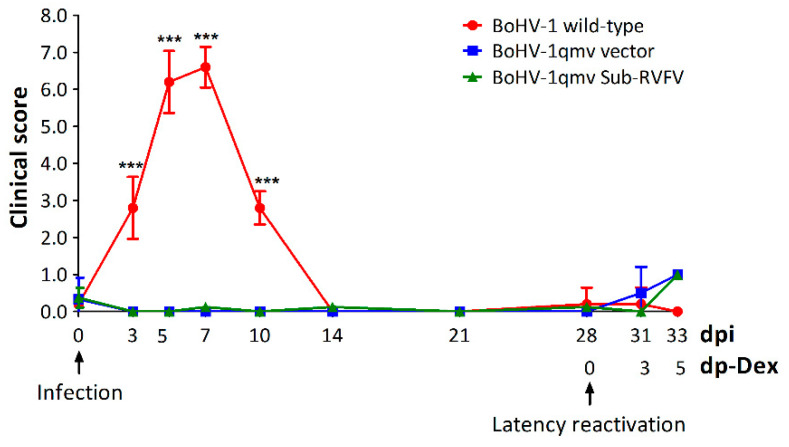
The clinical score in calves following bovine herpesvirus type 1 (BoHV-1) infection and latency reactivation. Criteria: rectal temperatures were scored 0–4 (<39.0 °C, 39.5 °C, 40.0 °C, 40.5 °C, and >40.9 °C), nasal discharges were scored 0–4 (normal, serous, mild, and severe mucopurulent), and nasal lesions were scored 0–3 (normal, hyperemia, pustules and ulcers). There was a significantly higher clinical score in the BoHV-1 wild-type group compared to the BoHV-1qmv vector and BoHV-1qmv Sub-RVFV vaccine group on days 5 and 7 post infection. We used two-way ANOVA followed by Bonferroni post tests to compare replicate means by row; *** *p* < 0.001. dpi—days post infection; dp-Dex—days post dexamethasone injection.

**Figure 3 viruses-16-01497-f003:**
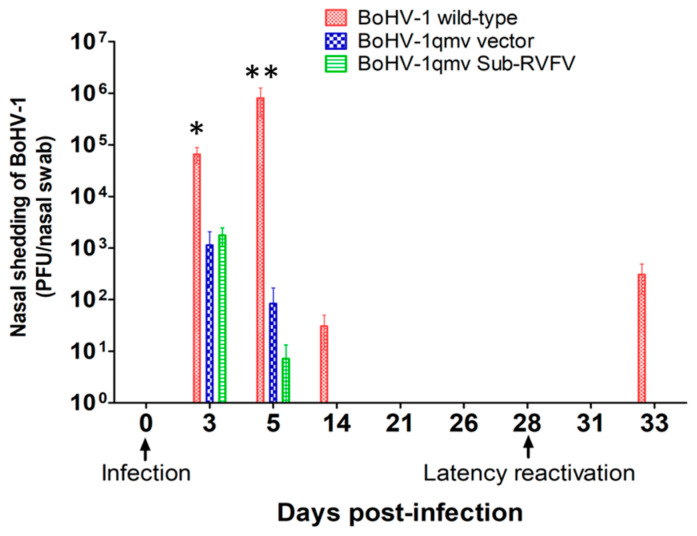
Nasal virus shedding following primary infection and after dexamethasone-induced latency re-activation determined by the virus plaque assays on MDBK cells. The Shapiro–Wilk test was performed to determine the normal distribution of the data. Normally distributed data were analyzed by two-way ANOVA followed by Bonferroni post tests to compare replicate means by row; * *p* < 0.05; ** *p* < 0.01.

**Figure 4 viruses-16-01497-f004:**
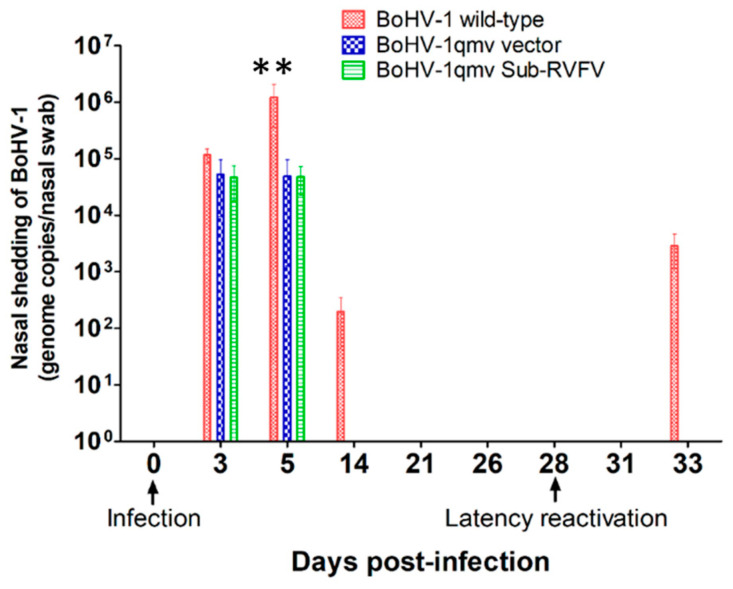
Nasal virus shedding following primary infection, during latency and upon dexamethasone-induced latency reactivation as determined by sensitive qPCR test using BoHV-1 major capsid protein (MCP)-specific probes. The detection limit was determined to be 10 copies per reaction, as evidenced by the ability to reliably detect the target at this concentration with a consistent amplification efficiency of 95% [25]. The assay demonstrated high reproducibility with a coefficient of variation of less than 5% across three independent runs. The Shapiro–Wilk test was performed to determine the normal distribution of the data. Normally distributed data were analyzed by two-way ANOVA followed by Bonferroni post tests to compare replicate means by row; ** *p* < 0.01.

**Figure 5 viruses-16-01497-f005:**
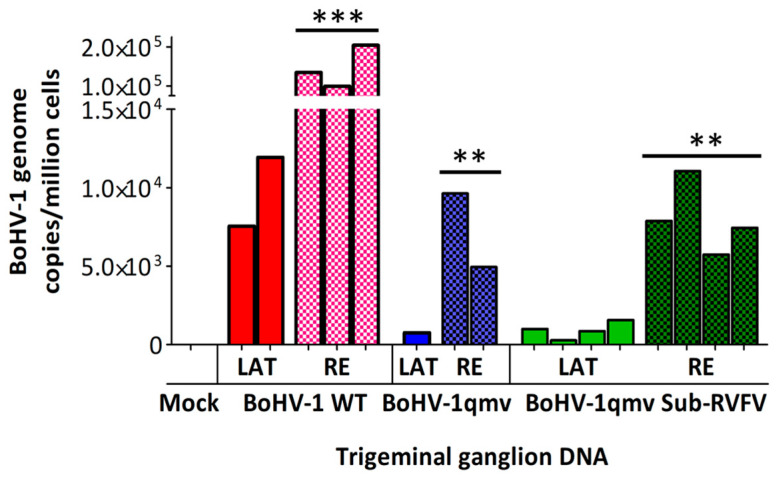
Quantifying BoHV-1 genome copies in the trigeminal ganglion (TG) of infected calves by BoHV-1 major capsid protein (MCP) gene-specific qPCR. Calves were infected with BoHV-1 wild-type (wt) or immunized with BoHV-1qmv vector or BoHV-1qmv Sub-RVFV vaccine. At 28 days post infection, latently infected calves were reactivated by dexamethasone treatment. At five days post reactivation, calves were euthanized, TGs were collected, and DNA was isolated from 25 mg of TG tissues. The qPCR was performed on isolated DNA samples targeting the BoHV-1-MCP gene. The BoHV-1 gene copies were calculated by normalization of the MCP-specific CT values against the standard curve generated based on the CT values obtained from the same samples for a cellular housekeeping GAPDH gene. The mean copy numbers of the BoHV-1-MCP gene copies per one million cells are shown. Two independent qPCR tests were performed for each TG sample. The bar graph represents the individual values in each group. The Shapiro–Wilk test was performed to determine the normal distribution of the data. Normally distributed data were analyzed by two-way ANOVA followed by Bonferroni post tests to compare replicate means by row; ** *p* < 0.01; *** *p* < 0.001. BoHV-1wt—BoHV-1 wt-infected calves (*n* = 5; 2—latency and 3—reactivated); BoHV-1qmv—BoHV-1qmv vector-infected calves (*n* = 3; 1—latency and 2—reactivated); BoHV-1qmv Sub-RVFV—BoHV-1qmv Sub-RVFV vaccine-infected calves (*n* = 8; 4—latency and 4—reactivated); Mock—TG from mock-infected calf; LAT—latency; RE—reactivated.

**Figure 6 viruses-16-01497-f006:**
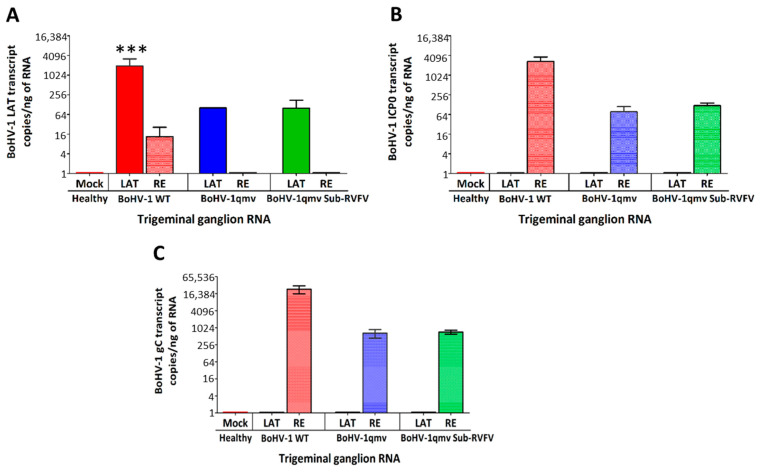
Quantifying BoHV-1-specific latency-related (LR), ICP0 and glycoprotein C gene transcript copies in infected calves’ trigeminal ganglion (TG). Calves were infected with BoHV-1 wt or immunized with BoHV-1qmv vector or BoHV-1qmv Sub-RVFV vaccine. At 28 days post infection, latently infected calves were reactivated by dexamethasone treatment. At five days post reactivation, calves were euthanized, and TGs were collected, and the total RNA was isolated from 25 mg of TG tissues. DNase-treated RNA was used for cDNA synthesis, and qPCR was performed targeting BoHV-1 (**A**) latency-related transcripts, (**B**) immediate early gene transcript ICP0, and (**C**) late protein transcript glycoprotein C. DNase-treated RNA without cDNA synthesis was included as a control to determine the efficacy of DNase treatment. Targeted transcript copies were calculated according to CT values of the standard curve. Calculated transcript copies were normalized and expressed per ng of RNA. Two independent qPCR analyses were performed for each calf. The bar graph represents the individual values in each group. The Shapiro–Wilk test was performed to determine the normal distribution of the data. Normally distributed data were analyzed by two-way ANOVA followed by Bonferroni post tests to compare replicate means by row; *** *p* < 0.001 between BoHV-1 wild-type LAT and BoHV-1 wild-type LAT-RE. BoHV-1 wt—BoHV-1 wt-infected calves (*n* = 5; 2—latency and 3—reactivated); BoHV-1qmv—BoHV-1qmv vector-infected calves (*n* = 3; 1—latency and 2—reactivated); BoHV-1qmv Sub-RVFV—BoHV-1qmv Sub-RVFV vaccine-immunized calves (*n* = 8; 4—latency and 4—reactivated); Mock—TG from mock-infected calves; LAT—latency; RE—reactivated.

**Figure 7 viruses-16-01497-f007:**
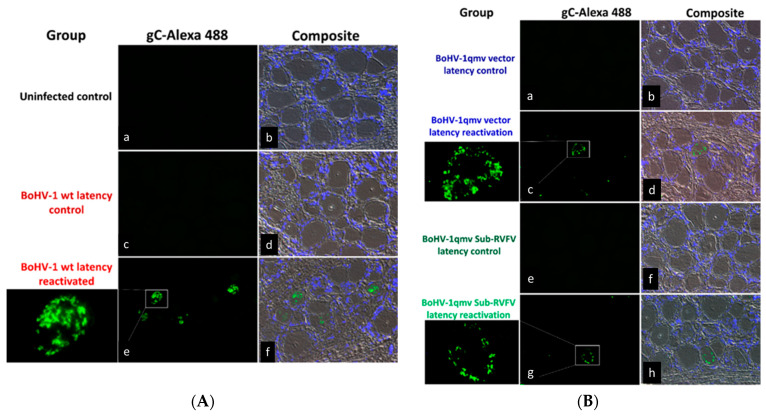
Bovine herpesvirus type 1 (BoHV-1) glycoprotein C expression in the trigeminal ganglion (TG) of the (**A**) BoHV-1- wild-type- (wt; **a**–**f**), (**B**) BoHV-1qmv vector- (**a**–**d**) and BoHV-1qmv Sub-RVFV- (**e**–**h**) infected calves determined by indirect immunofluorescence assay (IFA). Calves were infected with the respective viruses, and at 28 days post infection, latently infected calves were reactivated by dexamethasone (Dex) treatment. At five days post reactivation, calves were euthanized, TGs were collected, and formalin-fixed, and paraffin-embedded tissue sections were prepared. BoHV-1 gC-specific IIFA was performed in TG tissue sections collected from infected calves during latency or after reactivation. Bright apple-green fluorescent signals indicated positive signals.

**Figure 8 viruses-16-01497-f008:**
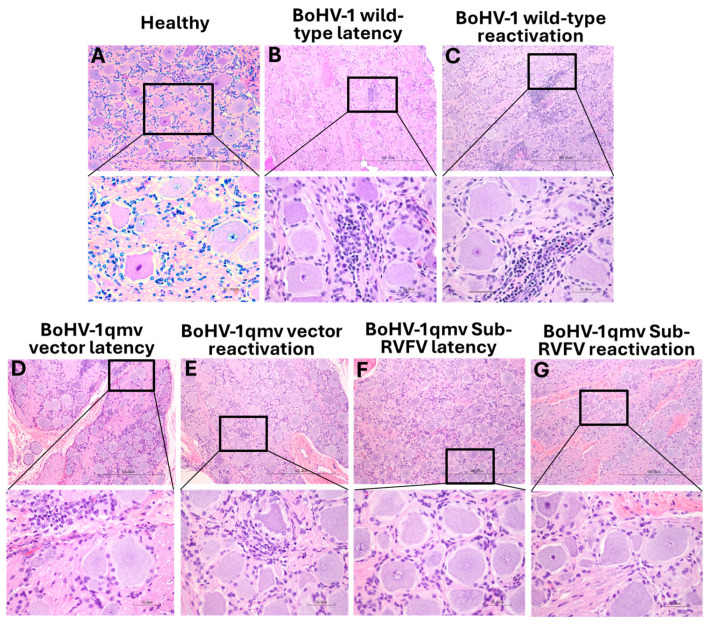
Histological analysis of trigeminal ganglion (TG) tissues collected from calves. Calves were infected with BoHV-1 wild-type (wt) or immunized with BoHV-1qmv vector/BoHV-1qmv Sub-RVFV vaccine virus. At 28 days post infection, latently infected calves were reactivated by dexamethasone injection. At five days post reactivation, calves were euthanized, TGs were collected, and formalin-fixed, and paraffin-embedded tissue sections were prepared and stained for histological analysis. (**A**) TG collected from the healthy calf revealed a normal morphology. A moderately large inflammatory focus was observed in the TG of calves infected with BoHV-1 wild-type following (**B**) latency and (**C**) latency reactivation. In contrast, no apparent inflammation was seen in both (**D**,**E**) BoHV-1qmv vector-immunized and (**F**,**G**) BoHV-1qmv Sub-RVFV-immunized calves during (**D** and **F**, respectively) latency and (**E** and **G**, respectively) after reactivation. Lower panel figures are enlarged sections of corresponding figures.

**Figure 9 viruses-16-01497-f009:**
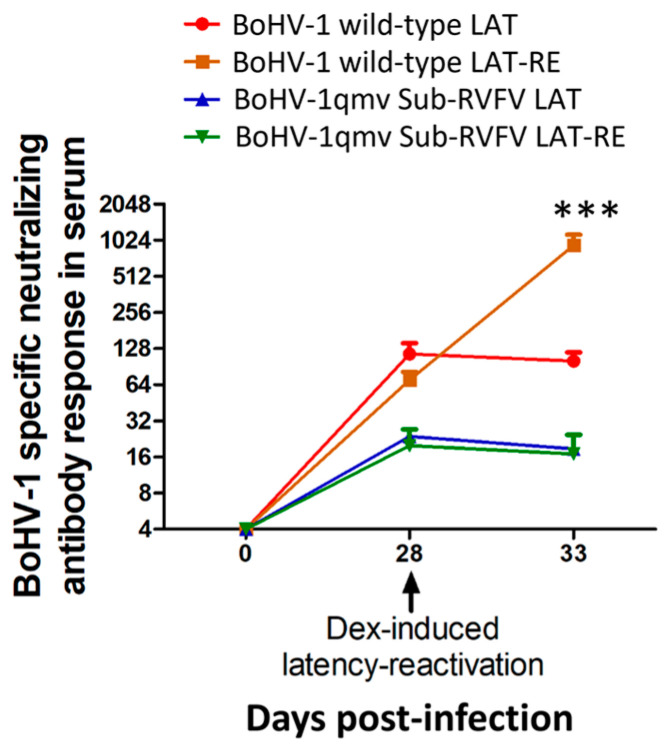
Bovine herpesvirus type 1 (BoHV-1)-specific serum neutralizing antibody (SN) titers developed in calves following primary infection, during latency, and after dexamethasone included latency reactivation. The data represent the mean ± standard deviation. The Shapiro–Wilk test was performed to determine the normal distribution of the data. Normally distributed data were analyzed by two-way ANOVA followed by Bonferroni post tests to compare replicate means by row; *** *p* < 0.001 between BoHV-1 wild-type LAT and BoHV-1 wild-type LAT-RE. (*n* = 2 for BoHV-1 wild-type latency group, *n* = 3 for BoHV-1 wild-type latency reactivation group, *n* = 4 for BoHV-1qmv Sub-RVFV latency group, and *n* = 4 for BoHV-1qmv Sub-RVFV latency reactivation group) LAT—latency; RE—reactivation.

**Table 1 viruses-16-01497-t001:** List of primers, probes, and double-stranded gene blocks (ds-gBlock as standard) used in quantitative PCR for quantification of bovine herpesvirus type 1 (BoHV-1) major capsid protein, ICP0, glycoprotein C, RVFV Gn, RVFV Gc and latency-related gene/transcript copies in TG tissue samples and subsequent normalization based on glyceraldehyde 3-phosphate dehydrogenase (GAPDH) housekeeping gene.

Primer/Probe/ds-gBlock	Name	Sequence
BoHV-1 major capsid protein	Forward	5′-TTTGGAGGCCCTAGAGAAGC-3′
	Reverse	5′-AAACGTCAGGTCCATGTTGC-3′
	Probe	5′FAM-CGGGTGCCCTACCCGCTGGT-3′TAMRA
	ds-gBlock	5′-CCGTTGGGGACCGGCTAGTGTTTTTGGAGGCCCTAGAGAAGCGC GTGTACCAGGCCACGCGGGTGCCCTACCCGCTGGTAGGCAACATGG ACCTGACGTTTGTCATGCCGCTGGGGCTGTACAAA-3′
BoHV-1 ICP0	Forward	5′-TGCTGACATACTGTCTTTCCGC-3′
	Reverse	5′-CTTGGTCGGAGTCAGAAGAGTC-3′
	Probe	5′FAM-CGCCTCGACTCAACCTCCGCGTCCGTCT-3′IB^®^FQ
	ds-gBlock	5′-TGTTTGTCGACCTCTAGTGCTGACATACTGTCTTTCCGCAGGCCCGGGC CGCGCCAGACGCCCCCCGCGTGCGCCTCGACTCAACCTCCGCGTCCGTCTCCGGCTCCGTTTCCGTGTCGTCAATGGCGGTCAGGTCGGAGGTGCTGAGCCCCGACGACGACTCTTCTGACTCCGACCAAGAGT-3′
BoHV-1 latency-related transcript	Forward	5′-ACAGACAGACAAAAAGCCAG-3′
	Reverse	5′-GTCATCCCCAAACCGAAAG-3′
	Probe	5′FAM - CATGCGCGACCTGGGCCATAA-3′IB^®^FQ
	ds-gBlock	5′-TCAGAGTTTAATATTACAGACAGACAAAAAGCCAGATAATTACAAAGTATTT GTTTTTATTGATTGCGCATGCGCGACCTGGGCCATAAAAGCCCCGCGCATG CGCGAGCAGTTACTTTCGGTTTGGGGATGACAGCGGCGACTGCGGCT-3′
BoHV-1 glycoprotein C	Forward	5′-ACCTTGCGCTGTGACTT-3′
	Reverse	5′-GAGCTGGGCGACAACTAT-3′
	Probe	5′FAM-ATGTCGCGGCGCCAAGTGTA-3′IB^®^FQ
	ds-gBlock	5′-GACGGTCACGACCTTGCGCTGTGACTTGGTGCCCATGTCGCGG CGCCAAGTGTACACGCCCTCGGTGGCGGCCGTCAGGGAGCGCACGGTCAGGGGCAAGTTGCGGGGGTCGGCGGGCGAAGGGAAAATATAGTTGTCGCCCAGCTCCGCGCTACGG-3′
RVFV Gn	Forward	5′-AGACCAAGAGGGAGCTGAAG-3′
	Reverse	5′-TCACCTGGATCTGCACGATG-3′
	Probe	5′FAM-CCAAGATCGGCGGCCACGGCTCCA-3′IB^®^FQ
RVFV Gc	Forward	5′-TGAAGGAGGACCAGACCAAG-3′
	Reverse	5′-CACGTTGAAACATCCGCAG-3′
	Probe	5′FAM-CGGGACAACGAGACCAGCGCCGAGTTCA-3′IB^®^FQ
Bovine glyceraldehyde 3-phosphate dehydrogenase (GAPDH)	Forward	5′-CATGACCACTTTGGCATCGT-3′
	Reverse	5′-CCATCCACAGTCTTCTGGGT-3′
	Probe	5′FAM-ACCACTGTCCACGCCATCACTGCC-3′TAMRA
	ds-gBlock	5′-GGCCCCCCTGGCCAAGGTCATCCATGACCACTTTGGCATCG TGGAGGGACTTATGACCACTGTCCACGCCATCACTGCCACCCAGAA GACTGTGGATGGCCCCTCCGGGAAGCTGTGGCGT-3′

## Data Availability

All relevant data for the study are included in the manuscript.

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
