# Peer review of "A Quadruple Gene-Deleted Live BoHV-1 Subunit RVFV Vaccine Vector Reactivates from Latency and Replicates in the TG Neurons of Calves but Is Not Transported to and Shed from Nasal Mucosa"

_viruses, 2024, doi:10.3390/v16091497_

Round 1
Reviewer 1 Report
Comments and Suggestions for Authors
In the manuscript by Pavulraj et al, the authors present a study evaluating the degree of attenuation of an attenuated bovine herpesvirus vaccine vector, recombinantly expressing the glycoproteins of Rift Valley fever virus. The manuscript represents a continuation of a previous paper by the authors (doi: 10.3390/v15112183) where the authors initially characterize their new RVFV vaccine, and demonstrate that it can induce protective immunity in calves. In this study, the authors compare both their recombinant RVFV vaccine, as well as the vaccine backbone (i.e without the RVFV cDNA) to that of wild-type BHV. The authors conclusively demonstrate the their quadruple mutant BHV is significantly attenuated, as evidenced by decreased dexamethasone-induced latency-reactivation, as well as a lack of virus-shedding in the nose of infected animals. The manuscript is suitable for publication, provided that the authors address the following concerns:
Major Concerns:
- the authors should not say call their vaccine a “subunit vaccine”. While the authors have also used this in their previous publication, it is not correct. What the authors have is a viral vector expressing recombinant RVFV glycoproteins. That is different from a subunit vaccine.
-the reviewer would like to know why the wt control animal was inoculated only intranasally, while the other experimental groups were inoculated both intranasally and subcutaneously
-considering the variability in internal temperatures in regular livestock, do you think having a clinical score based 0.5oC increments is a good idea?
Minor concerns:
Line 94-95: Please re-phrase (i.e. “we 94 combined gE CT and entire Us9 ORF deleted in the backbone of BoHV- 1 UL49.5 mutant...)
Line 102: “To improve further...” should be “To further improve...”
Line 107: “...and determined”
Line 477: “Because they each have simultaneous gE CT and Us9 deletions. “—This is not a complete sentence.
Final paragraph-- The authors need to be careful about rendering the virus completely replication-deficient. While the vector may be completely attenuated, the authors may also limit the extent of RVFV protein expression.
Comments on the Quality of English Language
Aside from a few awkwardly-written sentences, the English is fine.
Author Response
Reviewer #1
- the authors should not say call their vaccine a “subunit vaccine”. While the authors have also used this in their previous publication, it is not correct. What the authors have is a viral vector expressing recombinant RVFV glycoproteins. That is different from a subunit vaccine.
Response: We disagree. According to NIAID’s definition, subunit vaccines include only the components, or antigens, instead of the entire pathogen that best stimulates the immune system (https://www.niaid.nih.gov/research/vaccine-types ). We designated our vaccine as BoHV-1qmv vectored subunit RVFV vaccine (in short BoHV-1qmv Sub-RVFV), which is correct because it’s a live virus vectored subunit vaccine. Therefore, we insist that our designation of the vaccine is correct.
-the reviewer would like to know why the wt control animal was inoculated only intranasally, while the other experimental groups were inoculated both intranasally and subcutaneously
Response: Naturally, the primary WT virus infection occurs intranasally. Which is followed by latent infection in the TG. This manuscript characterized our BoHV-1qmv Sub-RVFV concerning latency-reactivation compared with the WT to demonstrate that the vaccine virus establishes latency in the TG but does not shed in the nose following reactivation. For the vaccine virus, we used both intranasal and subcutaneous route vaccination to replicate the vaccination route we used previously to demonstrate that regardless of the vaccination route, the latent virus in the TG (IN route) and the DRG (dorsal route ganglia), is not shed in the nose following DEX-induced reactivation.
-considering the variability in internal temperatures in regular livestock, do you think having a clinical score based 0.5oC increments is a good idea?
Yes.
Minor concerns:
Line 94-95: Please re-phrase (i.e. “we 94 combined gE CT and entire Us9 ORF deleted in the backbone of BoHV- 1 UL49.5 mutant...)
Done.
Line 102: “To improve further...” should be “To further improve...”
Done
Line 107: “...and determined”
Done.
Line 477: “Because they each have simultaneous gE CT and Us9 deletions. “—This is not a complete sentence.
Done
Final paragraph-- The authors need to be careful about rendering the virus completely replication-deficient. While the vector may be completely attenuated, the authors may also limit the extent of RVFV protein expression.
Response: TK is not necessary for the virus replication in the epithelial cells because the cellular TK complements the virus TK. The main purpose of the TK deletion is to make sure that the latent vaccine virus doesn’t replicate in the TG neurons because Neurons don’t have TK; they don’t divide (post-mitotic).
Reviewer 2 Report
Comments and Suggestions for Authors
Bovine Herpes Virus 1 (BoHV-1) infection occurs worldwide, and is an economically significant pathogen due to its impact on production losses. BoHV-1 cause life-long infection that reactivates under stress and may result in viral excretion. Modified attenuated variants of this virus can be used not only to control BoHV-1 infection, but also as a viral vector for the simultaneous prevention of several diseases. The authors created a BoHV-1 quadruple mutant virus (BoHV-1qmv) looks like a very promising viral vector for cattle immunization. The authors also note that they plan to modify this vector by deleting the TK gene. In my opinion, this is a good idea.
The authors have experience in developing vaccines based on various herpesvirus vectors. The article is well written, and the methodology used is adequate. The article is interesting and worthy of publication, however some minor issues need to be fixed:
1. The authors noted that statistical analyzes were performed, but there is no indication that the authors check if the distribution of the data met the assumptions of ANOVA? The p-values ​​for each comparative analysis in Figures 3–6 should be added. This will allow readers to assess the reliability of the conclusions.
2. The authors note that they used highly sensitive qPCR, but do not indicate its sensitivity. This information should be added.
3. There are typos in the text, for example, Line 501. The manuscript should be carefully checked.
Author Response
Reviewer #2
Bovine Herpes Virus 1 (BoHV-1) infection occurs worldwide, and is an economically significant pathogen due to its impact on production losses. BoHV-1 cause life-long infection that reactivates under stress and may result in viral excretion. Modified attenuated variants of this virus can be used not only to control BoHV-1 infection but also as a viral vector for the simultaneous prevention of several diseases. The authors created a BoHV-1 quadruple mutant virus (BoHV-1qmv) looks like a very promising viral vector for cattle immunization. The authors also note that they plan to modify this vector by deleting the TK gene. In my opinion, this is a good idea.
The authors have experience in developing vaccines based on various herpesvirus vectors. The article is well written, and the methodology used is adequate. The article is stimulating and worthy of publication, however some minor issues need to be fixed:
1. The authors noted that statistical analyses were performed, but there is no indication that the authors checked if the distribution of the data met the assumptions of ANOVA. The p-values ​​for each comparative analysis in Figures 3–6 should be added. This will allow readers to assess the reliability of the conclusions.
Response. P-values have been added.
- The authors note that they used highly sensitive qPCR, but do not indicate its sensitivity. This information should be added.
Response: The sensitivity limit has been incorporated in the Figure 4 legend (red fonts). Indicated.
3. There are typos in the text, for example, Line 501. The manuscript should be carefully checked.
Response: Revised.